# Combining CAR T-Cell Therapy and Nivolumab to Overcome Immune Resistance in THRLBCL: A Case Report

**DOI:** 10.3390/ijms26199265

**Published:** 2025-09-23

**Authors:** Daniel Munarriz, Oriana López-Godino, Nuria Martinez-Cibrian, Nil Albiol, Helena Brillembourg, Sergio Navarro-Velázquez, Marta Español-Rego, Sebastián Casanueva, Lucía García-Tomás, Guillermo Muñoz-Sanchez, Leticia Alserawan, Daniel Benitez-Ribas, Laura Magnano, Juan Gonzalo Correa, Andrea Rivero, Pablo Mozas, Eva Gine, Luis Gerardo Rodríguez-Lobato, Alexandra Martínez-Roca, Mercedes Montoro-Lorite, Pilar Ayora, Jordi Esteve, Laura Frutos, Olga Balagué-Ponz, Alvaro Urbano-Ispizua, Europa Azucena González-Navarro, Manel Juan, Julio Delgado, Valentín Ortiz-Maldonado

**Affiliations:** 1Department of Hematology, Hospital Clínic, 08036 Barcelona, Spain; munarriz@clinic.cat (D.M.); nmartinc@clinic.cat (N.M.-C.); nalbiol@clinic.cat (N.A.); brillembourg@recerca.clinic.cat (H.B.); lcmagnan@clinic.cat (L.M.); jgonzalo@clinic.cat (J.G.C.); anrivero@clinic.cat (A.R.); mozas@clinic.cat (P.M.); egine@clinic.cat (E.G.); lgrodriguez@clinic.cat (L.G.R.-L.); apmartinez@clinic.cat (A.M.-R.); mmontoro@clinic.cat (M.M.-L.); payora@clinic.cat (P.A.); jesteve@clinic.cat (J.E.); aurbano@clinic.cat (A.U.-I.); jdelgado@clinic.cat (J.D.); 2Department of Hematology, Centro Regional de Hemodonación, IMIB-Pascual Parrilla, Hospital Universitario Morales-Meseguer, 30008 Murcia, Spain; orilopezgodino@gmail.com (O.L.-G.); lucia.gt47@gmail.com (L.G.-T.); 3Faculty of Medicine, University of Barcelona, 08007 Barcelona, Spain; senavarro@clinic.cat (S.N.-V.); mjuan@clinic.cat (M.J.); 4Department of Immunology, Hospital Clínic, 08036 Barcelona, Spain; espanol@clinic.cat (M.E.-R.); gumunoz@clinic.cat (G.M.-S.); alserawan@clinic.cat (L.A.); dbenitezr@clinic.cat (D.B.-R.); eagonzal@clinic.cat (E.A.G.-N.); 5Institut d’Investigacions Biomèdiques August Pi i Sunyer (IDIBAPS), 08036 Barcelona, Spain; casanueva@clinic.cat (S.C.); obalague@clinic.cat (O.B.-P.); 6Department of Nuclear Medicine, Hospital C.U. Virgen de la Arrixaca, 30120 Murcia, Spain; laura.frutos@yahoo.es; 7Department of Pathology, Hospital Clínic de Barcelona, 08036 Barcelona, Spain; 8Josep Carreras Research Institute, 08916 Barcelona, Spain; 9Clínic-Sant Joan de Déu Immunotherapy Platform, 08036 Barcelona, Spain; 10Clinical Research Division, Fred Hutchinson Cancer Center, Seattle, WA 98109, USA

**Keywords:** THRLBCL, PD-1, nivolumab, CAR-T, CD19

## Abstract

T-cell/histiocyte-rich large B-cell lymphoma (THRLBCL) is a rare, aggressive subtype of diffuse large B-cell lymphoma characterized by a profoundly immunosuppressive tumor microenvironment. PD-L1 overexpression by tumor cells is a recognized immune escape mechanism and may underlie resistance to cellular therapies, including CAR T-cell therapy. We report a case of a 29-year-old woman with refractory stage IV-B THRLBCL treated with anti-CD19 CAR T-cell therapy (varnimcabtagene autoleucel), who achieved an initial response (day +28) but experienced disease progression by day +100 despite robust CAR T-cell expansion. Peripheral blood analysis revealed persistent absolute B-cell aplasia, while bone marrow biopsy confirmed CD19-positive disease. Comparative immunohistochemistry demonstrated markedly increased PD-L1 expression in post-CAR T-cell samples, suggesting adaptive immune resistance via PD-1/PD-L1-mediated CAR T-cell inhibition. Nivolumab was initiated at month +4 to overcome this checkpoint-mediated resistance. Notably, a complete metabolic response was documented on PET/CT after four doses of nivolumab (month +6). The patient remains in sustained remission, with persistent B-cell aplasia, four years post-intervention. This case provides clinical and pathological evidence supporting the use of immune checkpoint blockade to rescue CAR T-cell efficacy, highlighting the potential of this synergistic approach in THRLBCL and possibly other B-cell malignancies exhibiting similar immune evasion.

## 1. Introduction

T-cell/histiocyte-rich large B-cell lymphoma (THRLBCL) is a rare and distinct subtype of diffuse large B-cell lymphoma (DLBCL), characterized by a tumor microenvironment dominated by reactive T cells and histiocytes, with a relatively low proportion of malignant B cells. It shares histopathological and immunological features with nodular lymphocyte-predominant Hodgkin lymphoma (NLPHL), particularly regarding the pivotal role of immune modulation in its pathogenesis. Notably, high expression levels of programmed death ligands 1 and 2 (PD-L1 and PD-L2) have been documented in both entities, contributing to the suppression of anti-tumor immune responses and facilitating immune evasion mechanisms [1,2]. Indeed, checkpoint inhibitors targeting the PD1/PDL1 axis have been approved for the treatment of classical Hodgkin lymphoma (cHL) [3]. These therapeutic advances have opened the door to exploring their potential efficacy in other malignancies with similarly immunosuppressive tumor microenvironments, including THRLBCL [4].

## 2. Case Presentation

A 29-year-old woman was initially diagnosed with cHL (nodular sclerosis subtype 1) after presenting with one month of weight loss, night sweats, and painful inguinal lymphadenopathy. PET/CT revealed widespread supra- and infradiaphragmatic lymphadenopathy, with extensive spleen and extranodal involvement including bone, lungs, uterus and ovaries. She was initially treated with four cycles of ABVD chemotherapy, but an interim PET/CT reassessment revealed disease progression.

A second biopsy revealed a nodular lymphocyte predominant Hodgkin lymphoma (NLPHL) pattern. The patient subsequently received ESHAP as second-line therapy, without response. Treatment with brentuximab vedotin in combination with bendamustine was also ineffective. A new biopsy revised the diagnosis to THRLBCL including a high CD20 expression and a prominent T-cell infiltrate.

She was then treated sequentially with R-GEMOX, R-IFE, and R-CHOP as fourth-, fifth-, and sixth-line regimen, all without response. Given the lack of efficacy of available standard treatments and the patient’s preserved clinical status, she was considered eligible for compassionate-use CD19-targeted CAR T-cell therapy (varnimcabtagene autoleucel), following repeated confirmation of CD19 expression on tumor cells. Leukapheresis was performed, and R-CHOP plus dexamethasone was administered as bridging therapy. Pre-infusion PET/CT showed persistent supra- and infradiaphragmatic lymphadenopathy with extensive multifocal extranodal involvement in lungs and bone.

The patient received a fludarabine (30 mg/m^2^ for 3 days) plus cyclophosphamide (300 mg/m^2^ for 3 days) lymphodepleting chemotherapy followed by 5 × 10^6^ CAR-T cells/kg split in three fractions. Despite a high tumor burden, she experienced no cytokine release syndrome (CRS) or Immune effector cell-associated neurotoxicity syndrome (ICANS). PET/CT on day +28 showed a partial response, but disease progression was evident by day +100, despite robust CAR T-cell expansion and persistent B-cell aplasia (Figure 1).

A bone marrow biopsy confirmed a CD19-positive relapse, with 50% of tumor cells expressing CD19 surrounded by a dense T-cell infiltration. Immunohistochemistry revealed marked upregulation of PD-L1 in tumor cells, absent in pre-CAR T-cell samples (Figure 2), suggesting adaptive resistance via the PD-1/PD-L1 axis. These findings indicated impaired CAR T-cell efficacy within the tumor, with persistence of malignant CD19-positive THRLBCL cells (on-target, on-tumor refractoriness), despite clear evidence of CAR T-cell activity in peripheral blood through the absence of normal (non-malignant) CD19-positive B-cells (on-target, off-tumor effect). Therefore, nivolumab was proposed to overcome checkpoint-mediated inhibition.

Nivolumab monotherapy was started at month +4 post–CAR T-cell infusion. After four doses, PET/CT performed at month +6 revealed a complete metabolic response (Figure 3), and bone marrow biopsy confirmed complete remission. Two additional nivolumab doses were administered, but treatment was discontinued due to intercurrent prolonged COVID-19 infection and cryptogenic organizing pneumonia. As of the most recent follow-up, four years after CAR T-cell therapy, the patient remains relapse-free with ongoing B-cell aplasia and without further antitumor treatment. She maintains a good quality of life, with CD4/CD8 counts within the normal range and secondary hypogammaglobulinemia requiring periodic immunoglobulin replacement, without additional sequelae.

## 3. Discussion

This case highlights the critical role of the tumor microenvironment in the pathogenesis and therapeutic resistance of THRLBCL. Beyond modulating sensitivity to chemotherapy, the immunosuppressive microenvironment may also compromise the efficacy of novel cellular therapies, including CAR T-cell therapy. Several studies have described the underlying mechanisms by which malignant B-cells and histiocytes contribute to T-cell dysfunction in this context [5,6,7].

Programmed death-ligand 1 (PD-L1) is a well-characterized immune checkpoint molecule that regulates T-cell activity by suppressing cytokine secretion and inhibiting apoptosis of regulatory T cells (Tregs) [8]. In THRLBCL, PD-L1 is frequently overexpressed on malignant B cells, with PD-L1/PD-L2 gene amplification or copy number gains being among the most recurrent genomic alterations. As a result, tumor cells in THRLBCL display higher PD-L1 levels than those typically observed in conventional diffuse large B-cell lymphoma (DLBCL), resembling the immunophenotypic profile of Hodgkin Reed–Sternberg cells in classical Hodgkin lymphoma (cHL). Importantly, PD-L1 overexpression in THRLBCL is not limited to malignant B cells; tumor-associated macrophages (TAMs) also exhibit high PD-L1 expression—sometimes exceeding levels observed in cHL—thereby reinforcing local immunosuppression and promoting immune escape [1].

In addition, malignant B cells and TAMs contribute to other immunosuppressive mechanisms. Notably, overexpression of CCL8 and interferon-gamma (IFN-γ) has been reported, enhancing macrophage recruitment and activation, respectively. These activated macrophages produce immunosuppressive mediators such as indoleamine 2,3-dioxygenase (IDO) and V-set and immunoglobulin domain–containing protein 4 (VSIG4), which exert potent inhibitory effects on CD8^+^ T-cell function [9]. Moreover, IFN-γ has been shown to induce PD-L1 upregulation, further amplifying tumor immune evasion [10,11].

Collectively, these mechanisms contribute to a profoundly impaired anti-tumor immune response, likely underlying the aggressive clinical course and poor prognosis of THRLBCL. Although standard immunochemotherapy regimens—such as R-CHOP, R-CHOP/R-ICE, and dose-adjusted EPOCH—have shown some benefit, overall survival remains inferior compared to other DLBCL subtypes [12]. For patients with refractory disease, prognosis remains dismal despite multiple treatment lines. Consequently, immunotherapeutic approaches are increasingly being explored as strategies to overcome tumor immune evasion. In this context, biomarkers such as T-cell factor 1 (TCF1)—a transcription factor linked to stem-like CD8^+^ T cells—have emerged as potential predictors of favorable response to PD-1/PD-L1 blockade, and may help identify patients most likely to benefit [13].

CD19-directed CAR T-cell therapy has also demonstrated limited efficacy in THRLBCL, despite confirmed CD19 expression on malignant cells [4]. In line with these observations, Pophali et al. reported a 2-year progression-free survival rate of 29% in patients treated with the leading commercial anti-CD19 CAR T-cell products (axi-cel, tisa-cel, and liso-cel) as their sole treatment [14]. In the present case, complete remission was achieved only after the initiation of checkpoint inhibition, despite ongoing B-cell aplasia and robust CAR T-cell expansion. This finding supports the hypothesis that immune evasion mechanisms—mediated by PD-L1 and other immunosuppressive pathways—can directly impair CAR T-cell function.

To address this, PD-1 inhibitors such as nivolumab have been evaluated in the post–CAR T-cell setting. A study by Gazeau et al. investigated the addition of nivolumab after axicabtagene ciloleucel in patients with DLBCL and found that the benefit was limited to those who had achieved at least a partial response to CAR T-cell therapy. Patients with primary refractory disease did not respond to PD-1 blockade, and only 50% of those with partial responses converted to complete response—most of whom relapsed within one year. This limited efficacy contrasts with our case, in which a complete and durable response was achieved with a short course of nivolumab monotherapy. These differences may reflect the distinct biology of THRLBCL and underscore the importance of careful patient selection—particularly based on histopathologic features of checkpoint pathway upregulation—to identify candidates most likely to benefit from immune checkpoint inhibition [15]. Another possible consideration is the difference in co-stimulatory domains between CAR T-cell products. Axi-cel uses a CD28 domain, whereas varnimcabtagene autoleucel contains 4-1BB, which is often associated with a more memory-like and potentially less exhausted phenotype. Although speculative and without direct comparative evidence, such differences could have contributed to the distinct outcomes observed between the study by Gazeau et al. and our case.

Preliminary data also suggest the potential of combining anti–PD-L1 antibodies with CAR T-cell therapy in patients with THRLBCL, with reports of empirical use in small cohorts suggesting possible synergistic effects that warrant further investigation [16]. Our case provides both clinical and pathological evidence supporting the feasibility and efficacy of this combinatorial approach. Moreover, it demonstrates a durable and more precisely defined disease-free survival than that reported by Trujillano et al., and documents comorbidities that developed following immunotherapy. To our knowledge, this case represents the longest complete remission (>4 years) reported after sequential CD19-directed CAR T-cell therapy and PD-1 blockade in THRLBCL. These findings underscore the potential of checkpoint inhibition to restore CAR T-cell function in immune-evasive B-cell lymphomas such as THRLBCL.

## 4. Conclusions

THRLBCL is characterized by overexpression of the PD-1/PD-L1 axis, contributing to an immunomodulatory tumor microenvironment that underlies poor responses to current therapies, including CAR T-cell therapy. Our case provides evidence that CD19 CAR T-cell activity can be restored with checkpoint inhibition, such as nivolumab, in select patients in whom PD-1/PD-L1 overexpression is demonstrated.

## Figures and Tables

**Figure 1 ijms-26-09265-f001:**
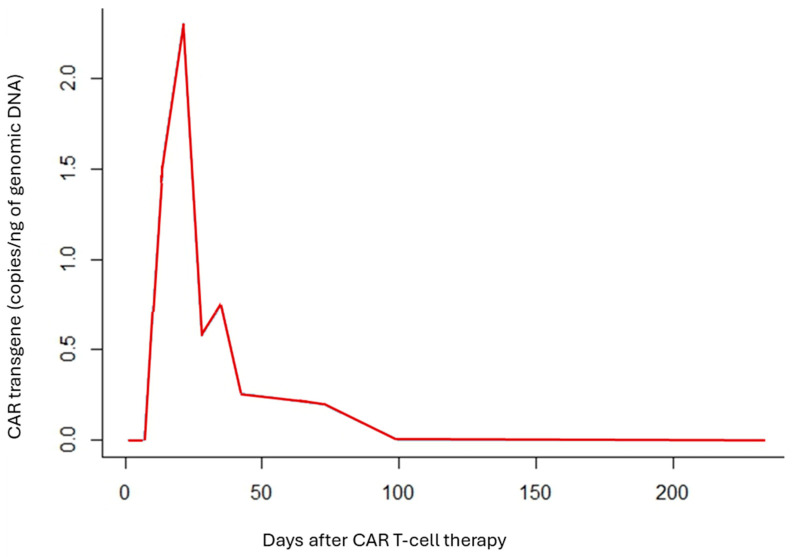
Overview of CAR-T chimera blood levels (Vector Copy Number; copies/cell) during the first 233 days after CART infusion. The patient experienced disease progression, as demonstrated by PET at day +100, despite detectable CAR T-cell chimerism in peripheral blood during this period. These findings suggest a role for tumor-associated immunomodulation in limiting CAR T-cell efficacy.

**Figure 2 ijms-26-09265-f002:**
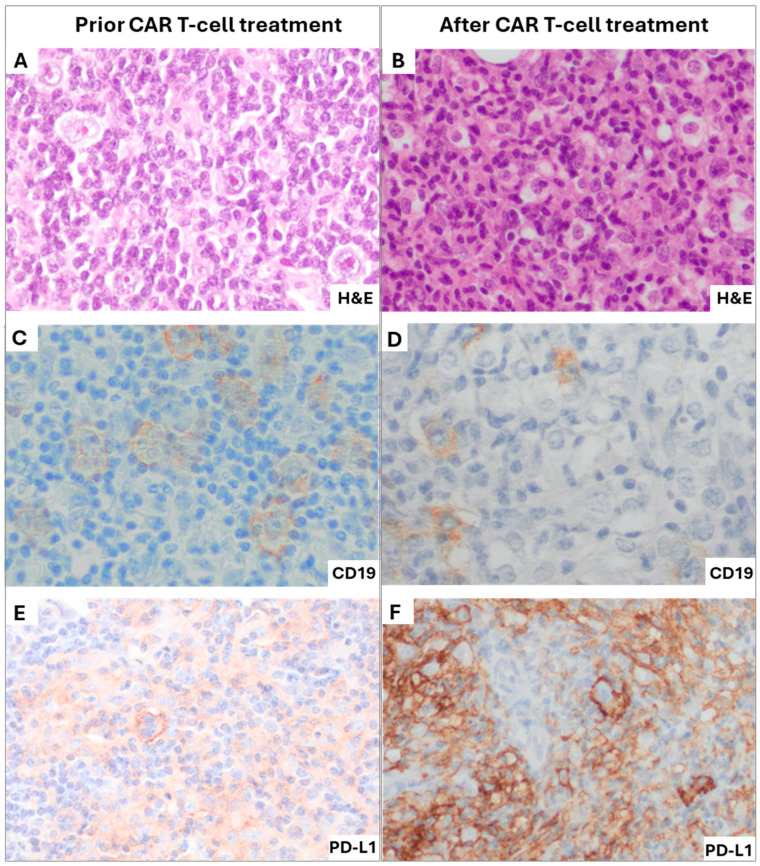
Histopathological features of the pre- and post-CAR T-cell therapy biopsies. The left column shows an adenopathy biopsy obtained prior to CAR T-cell infusion, while the right column shows a bone marrow biopsy obtained after CAR T-cell infusion. (**A**): H&E (40×) shows an infiltrate of large atypical cells with prominent nucleoli, scattered in a background of small mature lymphocytes and histiocytes; (**B**): H&E (40×) of the relapse in the bone marrow with a massive infiltrate showing large atypical B cells; (**C**): CD19 (40×) staining at diagnosis showed positivity in virtually all large tumoral B cells and in the small reactive B cells; (**D**): CD19 (40×) in bone marrow demonstrating retained expression in large tumoral B cells; (**E**): PD-L1 (40×) staining in the lymph node prior CAR T-cell therapy showed mild positivity in around 20% of large tumoral B-cells; (**F**): PD-L1 (40×) in the bone marrow showing strong upregulation and diffuse positivity in 80% of large tumoral B cells and in about 50% of the TME cells.

**Figure 3 ijms-26-09265-f003:**
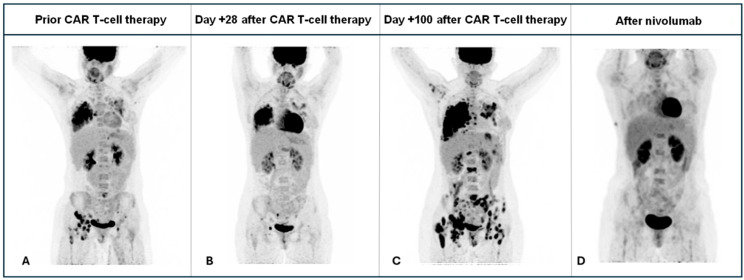
PET/CT-based disease assessment over time, shown at four key time points: before CAR T-cell infusion, at day +28, at day +100, and following nivolumab therapy. The panel displays coronal fused PET/CT images. (**A**) Baseline PET/CT prior to CAR T-cell infusion showing nodal and extranodal involvement, particularly in the right lung and pelvis. (**B**) PET/CT at day +28, demonstrating a partial response, most pronounced in pelvic lymphadenopathy. (**C**) PET/CT at day +100, revealing extensive disease progression. (**D**) PET/CT at day +175, after four doses of nivolumab, demonstrating a complete metabolic response (mCR).

## Data Availability

The original contributions presented in this study are included in the article. Further inquiries can be directed to the corresponding author(s).

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
