# Peer review of "Combining CAR T-Cell Therapy and Nivolumab to Overcome Immune Resistance in THRLBCL: A Case Report"

_ijms, 2025, doi:10.3390/ijms26199265_

Round 1

Reviewer 1 Report

Comments and Suggestions for Authors

This is a well-written and clinically significant case report describing a rare instance of T-cell/histiocyte-rich large B-cell lymphoma (THRLBCL) relapse after CAR T-cell therapy, with successful salvage using PD-1 blockade (nivolumab). The narrative is coherent, the clinical course is well documented, and the mechanistic link between PD-L1 upregulation and CAR T resistance is compelling. The sustained four-year remission adds substantial weight to the report’s impact.

This case has novelty value, potentially representing one of the first detailed clinical and pathological documentations of PD-1 blockade rescuing CAR T efficacy in THRLBCL.

The terms “on-target off-tumor effect” and “on-target on-tumor refractoriness” are used without prior definition. For readers outside the CAR T immunology field, brief clarifications are warranted in the abstract/manuscript.

The abstract would also benefit from a concise timeline summarizing the diagnosis, CAR T infusion, relapse, PD-L1 analysis, nivolumab administration, and follow-up duration. This would help readers follow the sequence of events more easily.

Finally, the manuscript should explicitly state whether this is the first reported THRLBCL case showing long-term remission after PD-1 rescue post-CAR T therapy. If similar cases exist, a brief literature comparison is warranted.

Author Response

Comment 1: The terms “on-target off-tumor effect” and “on-target on-tumor refractoriness” are used without prior definition. For readers outside the CAR T immunology field, brief clarifications are warranted in the abstract/manuscript.

Response 1: We sincerely thank you for this valuable comment. To address it, we have removed the terms “on-target off-tumor effect” and “on-target on-tumor refractoriness” from the Abstract, since the word limit does not allow for an adequate explanation. Instead, we have used these terms in the Case Presentation section (page 3, lines 95 and 97), where they can be properly contextualized.

Comment 2: The abstract would also benefit from a concise timeline summarizing the diagnosis, CAR T infusion, relapse, PD-L1 analysis, nivolumab administration, and follow-up duration. This would help readers follow the sequence of events more easily.

Response 2: We greatly appreciate this pertinent suggestion. To provide greater detail regarding the chronology of events described in the Abstract, we have incorporated the post–CAR T-cell infusion time points for each major milestone in the clinical course. This can be seen as follows: “who achieved an initial response (day +28) but experienced disease progression by day +100 despite robust CAR T-cell expansion (…) Nivolumab was initiated at month +4 (…) documented on PET/CT after four doses of nivolumab (month +6)” (Abstract, lines 19, 35, and 37). We believe this modification improves the clarity and readability of the sequence of events.

Comment 3: Finally, the manuscript should explicitly state whether this is the first reported THRLBCL case showing long-term remission after PD-1 rescue post-CAR T therapy. If similar cases exist, a brief literature comparison is warranted.

Response: We fully agree with this insightful comment. To the best of our knowledge, our case represents the longest well-documented response reported to date. To reflect this, we have incorporated the following statement into the Discussion: “To our knowledge, this case represents the longest complete remission (>4 years) reported after sequential CD19-directed CAR T-cell therapy and PD-1 blockade in THRLBCL” (page 7, lines 207–209).

Reviewer 2 Report

Comments and Suggestions for Authors

The case report presents a patient with refractory T-cell/histiocyte-rich  large B-cell lymphoma (THRLBCL) that reached remission after CD19-directed CAR T-cell therapy followed by PD-1 blocking with nivolumab. The manuscript is clearly written and methodologically sound. It provides a detailed clinical narrative supported by histopathological and immunohistochemical data as well as imaging. The correlation between PD-L1 upregulation and CAR-T cell therapy is supported by immunohistochemistry data. PET/CT biopsy data before/after intervention adds confidence to the manuscripts conclusions.

Some minor comments:

  1. The discussion would benefit from more comparisons with other case reports or trials involving checkpoint inhibitors post-CAR-T therapy in THRLBCL or DLBCL
  2. Histological panels need clearer legends and additional labelling.
  3. Quantification of PD-L1 expression should be added if present.
  4. The case report reports on sustained remission for 4 years. It would be good if data on the patients quality of life, immune status and any adverse effects during this time.

Author Response

Comments 1: The discussion would benefit from more comparisons with other case reports or trials involving checkpoint inhibitors post-CAR-T therapy in THRLBCL or DLBCL

Response 1: We thank you for pointing this out. To address it, we have added references to previous reports that further enrich the current experience in the treatment of this disease. These have been incorporated into the Discussion (page 7, lines 179–182 and 205–207).

Comments 2: Histological panels need clearer legends and additional labelling. 

Response 2: We fully agree with this comment. Accordingly, we have improved the quality of the images in Figure 2, provided microscopy at the same magnification, and optimized the placement of the legend. In addition, we have expanded the description of the images, including quantification of PD-L1 expression, among other details. These changes can be seen in Figure 2 (page 5).

Comments 3: Quantification of PD-L1 expression should be added if present.

Response 3: We sincerely thank you for pointing this out. In response, we have added the description of PD-L1 expression in both microscopic images: “E: PD-L1 (40×) staining in the lymph node prior to CAR T-cell therapy showed mild positivity in approximately 20% of large tumoral B cells; F: PD-L1 (40×) in the bone marrow showing strong upregulation and diffuse positivity in 80% of large tumoral B cells and in about 50% of the TME cells.”

Comments 4: The case report reports on sustained remission for 4 years. It would be good if data on the patients quality of life, immune status and any adverse effects during this time.

Response 4: This is indeed a very pertinent suggestion. Fortunately, the patient currently maintains a good quality of life without relevant sequelae. In addition, we have included her present immunological status, which shows hypogammaglobulinemia. These changes can be found on page 4, lines 106–108: “She maintains a good quality of life, with CD4/CD8 counts within the normal range and secondary hypogammaglobulinemia requiring periodic immunoglobulin replacement, without additional sequelae.”

Reviewer 3 Report

Comments and Suggestions for Authors

The manuscript by Munarriz et al. presents a case report providing clinical and pathological evidence supporting the combined use of CAR-T cells and Nivolumab to overcome immune resistance in a female patient. The study is interesting and has potential for publication after addressing the following concerns:

  1. The manuscript would benefit from a more detailed description of the methods and a clear, concise final conclusion to improve clarity and understanding.
  2. The authors should provide detailed explanations of all figures and state the specific conclusions that can be drawn from each. Please check the caption for Figure 3.
  3. The authors should provide results of the bone marrow biopsy following Nivolumab therapy to substantiate the reported remission.
  4. The case study should clearly indicate the patient’s condition before therapy and highlight the current treatment strategy aimed at overcoming immune resistance.

Addressing these points will strengthen the manuscript and enhance its clarity and impact.

Author Response

Comments 1: The manuscript would benefit from a more detailed description of the methods and a clear, concise final conclusion to improve clarity and understanding.

Response 1: We are most grateful for this pertinent comment. In response, we have improved the description of the therapeutic rationale (page 3, lines 93–98) and added a concluding statement at the end of the manuscript (page 8, lines 214–218): “THRLBCL is characterized by overexpression of the PD-1/PD-L1 axis, contributing to an immunomodulatory tumor microenvironment that underlies poor responses to current therapies, including CAR T-cell therapy. Our case provides evidence that CD19 CAR T-cell activity can be restored with checkpoint inhibition, such as nivolumab, in selected patients in whom PD-1/PD-L1 overexpression is demonstrated.”

Comments 2: The authors should provide detailed explanations of all figures and state the specific conclusions that can be drawn from each. Please check the caption for Figure 3.

Response 2: We sincerely appreciate this comment, as it allowed us to detect a typographical error in Figure 3 and improve the illustrative value of the images. In response, we have provided more detailed explanations across all figures (pages 4–6), enhanced the overall image quality, and optimized the legends. Specifically, in Figure 2, we have also standardized the magnification for greater clarity.

Comments 3: The authors should provide results of the bone marrow biopsy following Nivolumab therapy to substantiate the reported remission.

Response: We thank the reviewer for highlighting this point. In the Case Presentation section, we have clarified that the bone marrow biopsy performed after achieving a complete response post-nivolumab was found to be disease-free (page 3, line 103): "bone marrow biopsy was disease-free and confirmed complete remission".

Comments 4: The case study should clearly indicate the patient’s condition before therapy and highlight the current treatment strategy aimed at overcoming immune resistance.

Response 4: We fully agree with this comment, as CAR T-cell therapy requires that the patient maintain an adequate clinical status. To clarify this, we have specified in the Case Presentation section that the patient was eligible due to her preserved clinical status (page 3, line 76). Regarding the second comment, we have improved the explanation of the therapeutic approach that we consider appropriate, as detailed in the Conclusions (page 8, lines 215–219): “Our case provides evidence that CD19 CAR T-cell activity can be restored with checkpoint inhibition, such as nivolumab, in selected patients in whom PD-1/PD-L1 overexpression is demonstrated.”

Round 2

Reviewer 3 Report

Comments and Suggestions for Authors

The authors have adequately addressed the reviewers’ comments. The revised manuscript is satisfactory and can be considered for publication.

Author Response

We are deeply grateful for all the corrections provided by the reviewers, as well as the comments from the editor. We firmly believe that, thanks to their input, the manuscript has improved in both clarity and scientific interest.